# Loss Functions for Multiset Prediction

**Sean Welleck**[1,2]**, Zixin Yao**[1]**, Yu Gai**[1]**, Jialin Mao**[1]**, Zheng Zhang**[1]**, Kyunghyun Cho**[2,3]

[1]New York University Shanghai
[2]New York University
[3]CIFAR Azrieli Global Scholar
{wellecks,zixin.yao,yg1246,jialin.mao,zz,kyunghyun.cho}@nyu.edu

## Abstract

We study the problem of multiset prediction. The goal of multiset prediction is to train a predictor that maps an input to a multiset consisting of multiple items. Unlike existing problems in supervised learning, such as classification, ranking and sequence generation, there is no known order among items in a target multiset, and each item in the multiset may appear more than once, making this problem extremely challenging. In this paper, we propose a novel multiset loss function by viewing this problem from the perspective of sequential decision making. The proposed multiset loss function is empirically evaluated on two families of datasets, one synthetic and the other real, with varying levels of difficulty, against various baseline loss functions including reinforcement learning, sequence, and aggregated distribution matching loss functions. The experiments reveal the effectiveness of the proposed loss function over the others.

## 1   Introduction

A relatively less studied problem in machine learning, particularly supervised learning, is the problem of multiset prediction. The goal of this problem is to learn a mapping from an arbitrary input to a multiset[1] of items. This problem appears in a variety of contexts. For instance, in the context of high-energy physics, one of the important problems in a particle physics data analysis is to count how many physics objects, such as electrons, muons, photons, taus, and jets, are in a collision event [5]. In computer vision, object counting and automatic alt-text can be framed as multiset prediction [25, 12].

In multiset prediction, a learner is presented with an arbitrary input and the associated multiset of items. It is assumed that there is no predefined order among the items, and that there are no further annotations containing information about the relationship between the input and each of the items in the multiset. These properties make the problem of multiset prediction unique from other well-studied problems. It is different from sequence prediction, because there is no known order among the items. It is not a ranking problem, since each item may appear more than once. It cannot be transformed into classification, because the number of possible multisets grows exponentially with respect to the maximum multiset size.

In this paper, we view multiset prediction as a sequential decision making process. Under this view, the problem reduces to finding a policy that sequentially predicts one item at a time, while the outcome is still evaluated based on the aggregate multiset of the predicted items. We first propose an oracle policy that assigns non-zero probabilities only to prediction sequences that result exactly in the target, ground-truth multiset given an input. This oracle is optimal in the sense that its prediction never decreases the precision and recall regardless of previous predictions. That is, its decision is optimal in any state (i.e., prediction prefix). We then propose a novel *multiset loss* which minimizes

the KL divergence between the oracle policy and a parametrized policy at every point in a decision trajectory of the parametrized policy.

We compare the proposed multiset loss against an extensive set of baselines. They include a sequential loss with an arbitrary rank function, sequential loss with an input-dependent rank function, and an aggregated distribution matching loss and its one-step variant. We also test policy gradient, as was done in [25] recently for multiset prediction. Our evaluation is conducted on two sets of datasets with varying difficulties and properties. According to the experiments, we find that the proposed multiset loss outperforms all the other loss functions.

## 2 Multiset Prediction

A multiset prediction problem is a generalization of classification, where a target is not a single class but a multiset of classes. The goal is to find a mapping from an input $x$ to a multiset $\mathcal{Y} = \{y_1, \ldots, y_{|\mathcal{Y}|}\}$, where $y_k \in \mathcal{C}$. Some of the core properties of multiset prediction are; (1) the input $x$ is an arbitrary vector, (2) there is no predefined order among the items $y_i$ in the target multiset $\mathcal{Y}$, (3) the size of $\mathcal{Y}$ may vary depending on the input $x$, and (4) each item in the class set $\mathcal{C}$ may appear more than once in $\mathcal{Y}$. Formally, $\mathcal{Y}$ is a multiset $\mathcal{Y} = (\mu, \mathcal{C})$, where $\mu : \mathcal{C} \to \mathbb{N}$ gives the number of occurrences of each class $c \in \mathcal{C}$ in the multiset. See Appendix A for a further review of multisets.

As is typical in supervised learning, in multiset prediction a model $f_\theta(x)$ is trained on a dataset $\{(x_i, \mathcal{Y}_i)\}_{i=1}^N$, then evaluated on a separate test set $\{(x_i, \mathcal{Y}_i)\}_{i=1}^n$ using evaluation metrics $m(\cdot, \cdot)$ that compare the predicted and target multisets, i.e. $\frac{1}{n} \sum_{i=1}^n m(\hat{\mathcal{Y}}_i, \mathcal{Y}_i)$, where $\hat{\mathcal{Y}}_i = f_\theta(x_i)$ denotes a predicted multiset. For evaluation metrics we use exact match $\text{EM}(\hat{\mathcal{Y}}, \mathcal{Y}) = \mathbb{I}[\hat{\mathcal{Y}} = \mathcal{Y}]$, and the $F_1$ score. Refer to Appendix A for multiset definitions of exact match and $F_1$.

## 3 Related Problems in Supervised Learning

Variants of multiset prediction have been studied earlier. We now discuss a taxonomy of approaches in order to differentiate our proposal from previous work and define strong baselines.

### 3.1 Set Prediction

**Ranking** A ranking problem can be considered as learning a mapping from a pair of input $x$ and one of the items $c \in \mathcal{C}$ to its score $s(x, c)$. All the items in the class set are then sorted according to the score, and this sorted order determines the rank of each item. Taking the top-$K$ items from this sorted list results in a predicted set (e.g. [6]). Similarly to multiset prediction, the input $x$ is arbitrary, and the target is a set without any prespecific order. However, ranking differs from multiset prediction in that it is unable to handle multiple occurrences of a single item in the target set.

**Multi-label Classification via Binary Classification** Multi-label classification consists of learning a mapping from an input $x$ to a subset of classes identified as $\mathbf{y} \in \{0, 1\}^{|\mathcal{C}|}$. This problem can be reduced to $|\mathcal{C}|$ binary classification problems by learning a binary classifier for each possible class. Representative approaches include binary relevance, which assumes classes are conditionally independent, and probabilistic classifier chains which decompose the joint probability as $p(\mathbf{y}|x) = \prod_{c=1}^{|\mathcal{C}|} p(y_c|\mathbf{y}_{<c}, x)$ [3, 17, 22, 7]. Since each $p(y_c|\mathbf{y}_{<c}, x)$ models *binary* membership of a particular class, their predictions form a set $\hat{\mathbf{y}} \in \{0, 1\}^{|\mathcal{C}|}$ rather than a multiset $\hat{\mathbf{y}} \in \mathbb{N}^{|\mathcal{C}|}$.

### 3.2 Parallel Prediction

**Power Multiset Classification** A brute-force approach based on the Combination Method in multi-label classification [22, 17], is to transform the class set $C$ into a set $M(C)$ of all possible multisets, then train a multi-class classifier $\pi$ that maps an input $x$ to one of the elements in $M(C)$. However, the number of all possible multisets grows exponentially in the maximum size of a target multiset,[2] rendering this approach infeasible in practice.

**One Step Distribution Matching**  Instead of considering the target multiset as an actual multiset, one can convert it into a distribution over the class set, using each item's multiplicity. That is, we consider a target multiset $\mathcal{Y}$ as a set of samples from a single, underlying distribution $q^*$ over the class set $C$, empirically estimated as $q^*(c|x) = \frac{1}{|\mathcal{Y}|} \sum_{y \in \mathcal{Y}} I_{y=c}$, where $I.$ is an indicator function. A model then outputs a point $q_\theta(\cdot|x)$ in a $|C|$-dimensional simplex and is trained by minimizing a divergence between $q_\theta(\cdot|x)$ and $q^*(c|x)$. The model also predicts the size $\hat{l}_\theta(x)$ of the target multiset, so that each unique $c \in C$ has a predicted cardinality $\hat{\mu}(c) = \text{round}(q_\theta^c(x) \cdot \hat{l}(x))$. An un-normalized variant could directly regress the cardinality of each class.

A major weakness of these methods is the lack of modeling dependencies among the items in the predicted multiset, a known issue in multi-label classification [3, 14]. We test this approach in the experiments ($\mathcal{L}_{\text{1-step}}$) and observe substantially worse prediction accuracy than other baselines.

## 3.3   Sequential Methods

**Sequence prediction**  A sequence prediction problem is characterized as finding a mapping from an input $x$ to a sequence of classes $\mathcal{Y}_{\text{seq}} = (y_1, ..., y_{|\mathcal{Y}|})$. It is different from multiset prediction since a sequence has a predetermined order of items, while a multiset is an unordered collection. Multiset prediction can however be treated as sequence prediction by defining an ordering for each multiset. Each target multiset $\mathcal{Y}$ is then transformed into an ordered sequence $\mathcal{Y}_{\text{seq}} = (y_1, ..., y_{|\mathcal{Y}|})$, a model predicts a sequence $\hat{\mathcal{Y}}_{\text{seq}} = (\hat{y}_1, ..., \hat{y}_{|\mathcal{Y}|})$, and a per-step loss $\mathcal{L}_{seq}$ is minimized using $\mathcal{Y}_{\text{seq}}$ and $\hat{\mathcal{Y}}_{\text{seq}}$.

Recently, multi-label classification (i.e. set prediction) was posed as sequence prediction with RNNs [24, 14], improving upon methods that do not model conditional label dependencies. However, these approaches and the $\mathcal{L}_{seq}$ approach outlined above require a pre-specified rank function which orders output sequences (e.g. class prevalence in [24]).

Because multiset prediction does not come with such a rank function by definition, we must choose a (often ad-hoc) rank function, and performance can significantly vary based on the choice. Vinyals et al. [23] observed this variation in sequence-based set prediction (also observed in [14, 24]), which we confirm for multisets in section 5.3. This shows the importance of our proposed method, which does not require a fixed label ordering.

Unlike $\mathcal{L}_{\text{seq}}$, our multiset loss $\mathcal{L}_{\text{multiset}}$ proposed below is permutation invariant with respect to the order of the target multiset, and is thus not susceptible to performance variations from choosing a rank function, since such a choice is not required. We use $\mathcal{L}_{\text{seq}}$ as a baseline in Experiment 3, finding that it underperforms the proposed $\mathcal{L}_{\text{multiset}}$ .

**Aggregated Distribution Matching**  As in one-step distribution matching, a multiset is treated as a distribution $q_*$ over classes. The sequential variant predicts a sequence of classes $(y_1, ..., y_{|\mathcal{Y}|})$ by sampling from a predicted distribution $q_\theta^{(t)}(y_t|y_{<t}, x)$ at each step $t$. The per-step distributions $q_\theta^{(t)}$ are averaged into an aggregate distribution $q_\theta$, and a divergence between $q_*$ and $q_\theta$ is minimized. We test $L_1$ distance and KL-divergence in the experiments ($\mathcal{L}_{\text{dm}}^p, \mathcal{L}_{\text{dm}}^{\text{KL}}$).

A major issue with this approach is that it may assign non-zero probability to an incorrect sequence of predictions due to the aggregated distribution's invariance to the order of predictions. This is reflected in an increase in the entropy of $q_\theta^{(t)}$ over time, discussed in Experiment 3.

**Reinforcement Learning**  In [25], an approach based on reinforcement learning (RL) was proposed for multiset prediction. In this approach, a policy $\pi_\theta$ samples a multiset as a sequential trajectory, and the goal is finding $\pi_\theta$ whose trajectories maximize a reward function designed specifically for multiset prediction. REINFORCE [26] is used to minimize the resulting loss function, which is known to be difficult due to high variance [16]. We test the RL method in the experiments ($\mathcal{L}_{\text{RL}}$).

## 3.4   Domain-Specific Methods

In computer vision, object counting and object detection are instances of multiset prediction. Typical object counting approaches in computer vision, e.g. [12, 28, 15], model the counting problem as density estimation over image space, and assume that each object is annotated with a dot specifying its location. Object detection methods (e.g. [21, 19, 18, 8]) also require object location annotations. Since these approaches exploit the fact the input is an image and rely on additional annotated

information, they are not directly comparable to our method which only assumes annotated class labels and is agnostic to the input modality.

# 4 Multiset Loss Function for Multiset Prediction

In this paper, we propose a novel loss function, called *multiset loss*, for the problem of multiset prediction. This loss function is best motivated by treating the multiset prediction problem as a sequential decision making process with a model being considered a policy $\pi$. This policy, parametrized by $\theta$, takes as input the input $x$ and all the previously predicted classes $\hat{y}_{<t}$ at time $t$, and outputs the distribution over the next class to be predicted. That is, $\pi_\theta(y_t|\hat{y}_{<t}, x)$.

We first define a free label multiset at time $t$, which contains all the items that remain to be predicted after $t-1$ predictions by the policy, as

**Definition 1** (Free Label Multiset).

$$\mathcal{Y}_t \leftarrow \mathcal{Y}_{t-1} \backslash \{\hat{y}_{t-1}\},$$

where $\hat{y}_{t-1}$ is the prediction made by the policy at time $t-1$.

We then construct an oracle policy $\pi_*$. This oracle policy takes as input a sequence of predicted labels $\hat{y}_{<t}$, the input $x$, and the free label multiset with respect to its predictions, $\mathcal{Y}_t = \mathcal{Y} \backslash \{\hat{y}_{<t}\}$. It outputs a distribution whose entire probability (1) is evenly distributed over all the items in the free label multiset $\mathcal{Y}_t$. In other words,

**Definition 2** (Oracle).

$$\pi_*(y_t|\hat{y}_{<t}, x, \mathcal{Y}_t) = \begin{cases} \frac{1}{|\mathcal{Y}_t|}, & \text{if } y_t \in \mathcal{Y}_t \\ 0, & \text{otherwise} \end{cases}$$

An interesting and important property of this oracle is that it is optimal given any prefix $\hat{y}_{<t}$ with respect to both precision and recall. This is intuitively clear by noticing that the oracle policy allows only a correct item to be selected. We call this property the optimality of the oracle.

**Remark 1.** Given an arbitrary prefix $\hat{y}_{<t}$,

$$\text{Prec}(\hat{y}_{<t}, \mathcal{Y}) \leq \text{Prec}(\hat{y}_{<t} \cup \hat{y}, \mathcal{Y}) \text{ and } \text{Rec}(\hat{y}_{<t}, \mathcal{Y}) \leq \text{Rec}(\hat{y}_{<t} \cup \hat{y}, \mathcal{Y}),$$

for any $\hat{y} \sim \pi_*(\hat{y}_{<t}, x, \mathcal{Y}_t)$.

From the remark above, it follows that the oracle policy is optimal in terms of precision and recall.

**Remark 2.**

$$\text{Prec}(\hat{y}_{\leq|\mathcal{Y}|}, \mathcal{Y}) = 1 \text{ and } \text{Rec}(\hat{y}_{\leq|\mathcal{Y}|}, \mathcal{Y}) = 1, \text{ for all } \hat{y}_{\leq|\mathcal{Y}|} \sim \prod_{t=1}^{|\mathcal{Y}|} \pi_*(y_t|y_{<t}, x, \mathcal{Y}_t).$$

It is trivial to show that sampling from such an oracle policy would never result in an incorrect prediction. That is, this oracle policy assigns zero probability to any sequence of predictions that is not a permutation of the target multiset.

**Remark 3.**

$$\prod_{t=1}^{|\mathcal{Y}|} \pi_*(y_t|y_{<t}, x) = 0, \text{ if multiset}(y_1, \ldots, y_{|\mathcal{Y}|}) \neq \mathcal{Y},$$

where multiset equality refers to exact match, as defined in Appendix 1.

In short, this oracle policy tells us at each time step $t$ which of all the items in the class set $C$ must be selected. By selecting an item according to the oracle, the free label multiset decreases in size. Since the oracle distributes equal probability over items in the free label multiset, the oracle policy's entropy decreases over time.

**Remark 4** (Decreasing Entropy).

$$\mathcal{H}(\pi_*^{(t)}) > \mathcal{H}(\pi_*^{(t+1)}),$$

where $\mathcal{H}(\pi_*^{(t)})$ denotes the Shannon entropy of the oracle policy at time $t$, $\pi_*(y|\hat{y}_{<t}, x, \mathcal{Y}_t)$.

Proofs of the remarks above can be found in Appendix B–D.

The oracle's optimality allows us to consider a step-wise loss between a parametrized policy $\pi_\theta$ and the oracle policy $\pi_*$, because the oracle policy provides us with an optimal decision regardless of the quality of the prefix generated so far. We thus propose to minimize the KL divergence from the oracle policy to the parametrized policy at each step separately. This divergence is defined as

$$\mathrm{KL}(\pi_*^t \| \pi_\theta^t) = \underbrace{\mathcal{H}(\pi_*^t)}_{\text{const. w.r.t. } \theta} - \sum_{y_j \in |\mathcal{Y}_t|} \frac{1}{|\mathcal{Y}_t|} \log \pi_\theta(y_j | \hat{y}_{<t}, x),$$

where $\mathcal{Y}_t$ is formed using predictions $\hat{y}_{<t}$ from $\pi_\theta$, and $\mathcal{H}(\pi_*^t)$ is the entropy of the oracle policy at time step $t$. This entropy term can be safely ignored when learning $\pi_\theta$, since it is constant with respect to $\theta$. We then define a per-step loss function as $\mathcal{L}^t(x, \mathcal{Y}, \hat{y}_{<t}, \theta) = \mathrm{KL}(\pi_*^t \| \pi_\theta^t) - \mathcal{H}(\pi_*^t)$. The KL divergence may be replaced with another divergence.

It is intractable to minimize this per-step loss for every possible state $(\hat{y}_{<t}, x)$, since the size of the state space grows exponentially with respect to the size of a target multiset. We thus propose here to minimize the per-step loss only for the state, defined as a pair of the input $x$ and the prefix $\hat{y}_{<t}$, visited by the parametrized policy $\pi_\theta$. That is, we generate an entire trajectory $(\hat{y}_1, \ldots, \hat{y}_T)$ by executing the parametrized policy until either all the items in the target multiset have been predicted or the predefined maximum number of steps have passed. Then, we compute the loss function at each time $t$ based on $(x, \hat{y}_{<t})$, for all $t = 1, \ldots, T$. The final loss function is the sum of all these per-step loss functions:

**Definition 3** (Multiset Loss Function).

$$\mathcal{L}_{\mathrm{multi}}(x, \mathcal{Y}, \theta) = -\sum_{t=1}^{T} \frac{1}{|\mathcal{Y}_t|} \sum_{y_j \in \mathcal{Y}_t} \log \pi_\theta(y_j | \hat{y}_{<t}, x),$$

where $T$ is the smaller of the smallest $t$ for which $\mathcal{Y}_t = \emptyset$ and the predefined maximum value.

By Remarks 2 and 3, minimizing this loss function maximizes F1 and exact match.

**Execution Strategies**   As was shown in [20], the use of the parametrized policy $\pi_\theta$ instead of the oracle policy $\pi_*$ allows the upper bound on the learned policy's error to be linear with respect to the size of the target multiset. If the oracle policy had been used, the upper bound would have grown quadratically with respect to the size of the target multiset. To confirm this empirically, we test the following three alternative strategies for executing the parametrized policy $\pi_\theta$: (1) Greedy search: $\hat{y}_t = \arg\max_y \log \pi_\theta(y | \hat{y}_{<t}, x)$, (2) Stochastic sampling: $\hat{y}_t \sim \pi_\theta(y | \hat{y}_{<t}, x)$, and (3) Oracle sampling: $\hat{y}_t \sim \pi_*(y | \hat{y}_{<t}, x, \mathcal{Y}_t)$. After training, the learned policy is evaluated by greedily selecting each item from the policy.

**Variable-Sized Multisets**   In order to predict variable-sized multisets with the proposed loss functions, we introduce a **termination policy** $\pi_s$, which outputs a stop distribution given the predicted sequence of items $\hat{y}_{<t}$ and the input $x$. Because the size of the target multiset is known during training, we simply train this termination policy in a supervised way using a binary cross-entropy loss. At evaluation time, we simply threshold the predicted stop probability at a predefined threshold (0.5). An alternative method for supporting variable-sized multisets is discussed in Appendix E.

**Relation to Learning to Search**   Our framing of multiset prediction as a sequential task based on learning to imitate an oracle policy is inspired by the Learning to Search (L2S) approach to structured prediction [2, 1]. Recently, Leblond et al. [10] proposed SeaRNN, adapting L2S to modern recurrent models. Our proposal can be seen as designing an oracle and loss with favorable properties for multiset prediction, using a learned roll-in $\pi_\theta$, and directly setting a cost vector equal to the oracle's distribution, avoiding the expensive per-step roll-out in SeaRNN. We believe that applying the general L2S framework to novel problem settings is an important research direction.

## 5   Experiments and Analysis

### 5.1   Datasets

**MNIST Multi**   MNIST Multi is a class of synthetic datasets. Each dataset consists of multiple 100x100 images, each of which contains a varying number of digits from the original MNIST [11].

| | Table 1: Influence of rank function choice | | | | Table 2: Execution Strategies | | |
|---|---|---|---|---|---|---|---|

| | **MNIST Multi (4)** | | **COCO Easy** | | | **COCO Medium** | |
| | EM | F1 | EM | F1 | | EM | F1 |
|---|---|---|---|---|---|---|---|
| **Random** | 0.920 | 0.977 | 0.721 | 0.779 | **Greedy** | $0.475 \pm 0.006$ | $0.645 \pm 0.016$ |
| **Area** | 0.529 | 0.830 | 0.700 | 0.763 | **Stochastic** | $0.475 \pm 0.004$ | $0.649 \pm 0.009$ |
| **Spatial** | 0.917 | 0.976 | 0.675 | 0.738 | **Oracle** | $0.469 \pm 0.002$ | $0.616 \pm 0.009$ |

We vary the size of each digit and also add clutter. In the experiments, we consider the following variants of MNIST Multi:

- **MNIST Multi (4)**: $|\mathcal{Y}| = 4$; 20-50 px digits
- **MNIST Multi (1-4)**: $|\mathcal{Y}| \in 1, \ldots, 4$; 20-50 px digits
- **MNIST Multi (10)**: $|\mathcal{Y}| = 10$; 20 px digits

Each dataset has a training set with 70,000 examples and a test set with 10,000 examples. We randomly sample 7,000 examples from the training set to use as a validation set, and train with the remaining 63,000 examples.

**MS COCO**  As a real-world dataset, we use Microsoft COCO [13] which includes natural images with multiple objects. Compared to MNIST Multi, each image in MS COCO has objects of more varying sizes and shapes, and there is a large variation in the number of object instances per image which spans from 1 to 91. The problem is made even more challenging with many overlapping and occluded objects. To better control the difficulty, we create the following two variants:

- **COCO Easy**: $|\mathcal{Y}| = 2$; 10,230 examples, 24 classes
- **COCO Medium**: $|\mathcal{Y}| \in 1, \ldots, 4$; 44,121 training examples, 23 classes

In both of the variants, we only include images whose $|\mathcal{Y}|$ objects are large and of common classes. An object is defined to be large if the object's area is above the 40-th percentile across the training set of MS COCO. After reducing the dataset to have $|\mathcal{Y}|$ large objects per image, we remove images containing only objects of rare classes. A class is considered rare if its frequency is less than $\frac{1}{|C|}$, where $C$ is the class set. These two stages ensure that only images with a proper number of large objects are kept. We do not use fine-grained annotation (pixel-level segmentation and bounding boxes) except for creating input-dependent rank functions for the $\mathcal{L}_{\text{seq}}$ baseline (see Appendix F.2).

For each variant, we hold out a randomly sampled 15% of the training examples as a validation set. We form separate test sets by applying the same filters to the COCO validation set. The test set sizes are 5,107 for COCO Easy and 21,944 for COCO Medium.

## 5.2  Models

**MNIST Multi**  We use three convolutional layers of channel sizes 10, 10 and 32, followed by a convolutional long short-term memory (LSTM) layer [27]. At each step, the feature map from the convolutional LSTM layer is average-pooled spatially and fed to a softmax classifier. In the case of the one-step variant of aggregate distribution matching, the LSTM layer is skipped.

**MS COCO**  We use a ResNet-34 [9] pretrained on ImageNet [4] as a feature extractor. The final feature map from this ResNet-34 is fed to a convolutional LSTM layer, as described for MNIST Multi above. We do not finetune the ResNet-34 based feature extractor.

In all experiments, for predicting variable-sized multisets we use the termination policy approach since it is easily applicable to all of the baselines, thus ensuring a fair comparison. Conversely, it is unclear how to extend the special class approach to the distribution matching baselines.

**Training and evaluation**  For each loss, a model was trained for 200 epochs (350 for MNIST Multi 10). After each epoch, exact match was computed on the validation set. The model with the highest validation exact match was used for evaluation on the test set. See Appendix E for more details.

When evaluating a trained policy, we use greedy decoding. Each predicted multiset is compared against the ground-truth target multiset, and we report both the exact match accuracy (EM) and F-1 score (F1), as defined in Appendix 1.

Table 3: Loss function comparison

| | (a) MNIST Variants | | | | | | (b) MS COCO Variants | | | |
| | Multi (4) | | Multi (1-4) | | Multi (10) | | Easy | | Medium | |
| | EM | F1 | EM | F1 | EM | F1 | EM | F1 | EM | F1 |
|---|---|---|---|---|---|---|---|---|---|---|
| $\mathcal{L}_{\text{multi}}$ | **0.950** | **0.987** | **0.953** | **0.981** | **0.920** | **0.992** | 0.702 | **0.788** | **0.481** | **0.639** |
| $\mathcal{L}_{\text{RL}}$ | 0.912 | 0.977 | 0.945 | 0.980 | 0.665 | 0.970 | 0.672 | 0.746 | 0.425 | 0.564 |
| $\mathcal{L}_{\text{dm}}^{1}$ | 0.921 | 0.978 | 0.918 | 0.969 | 0.239 | 0.714 | 0.533 | 0.614 | 0.221 | 0.085 |
| $\mathcal{L}_{\text{dm}}^{\text{KL}}$ | 0.908 | 0.974 | 0.908 | 0.962 | 0.256 | 0.874 | **0.714** | 0.763 | 0.444 | 0.591 |
| $\mathcal{L}_{\text{seq}}$ | 0.906 | 0.973 | 0.891 | 0.952 | 0.592 | 0.946 | 0.709 | 0.774 | 0.457 | 0.592 |
| $\mathcal{L}_{\text{1-step}}$ | 0.210 | 0.676 | 0.055 | 0.598 | 0.032 | 0.854 | 0.552 | 0.664 | 0.000 | 0.446 |

## 5.3 Experiment 1: Influence of a Rank Function on Sequence Prediction

First, we investigate the sequence loss function $\mathcal{L}_{\text{seq}}$ from Sec. 3.3, while varying a rank function. We test three alternatives: a random rank function[3] and two input-dependent rank functions $r_{\text{spatial}}$ and $r_{\text{area}}$. $r_{\text{spatial}}$ orders labels in left-to-right, top-to-bottom order, and $r_{\text{area}}$ orders labels by decreasing object area; see Appendix F for more detail. We compare these rank functions on MNIST Multi (4) and COCO Easy validation sets.

We present the results in Table 1. It is clear from the results that the performance of the sequence prediction loss function is dependent on the choice of a rank function. In the case of MNIST Multi, the area-based rank function was far worse than the other choices. However, this was not true on COCO Easy, where the spatial rank function was worst among the three. In both cases, we have observed that the random rank function performed best, and from here on, we use the random rank function in the remaining experiments. This set of experiments firmly suggests the need of an order-invariant multiset loss function, such as the proposed multiset loss function.

## 5.4 Experiment 2: Execution Strategies for the Multiset Loss Function

In this set of experiments, we compare the three execution strategies for the proposed multiset loss function, illustrated in Sec. 3. They are **greedy** decoding, **stochastic** sampling and **oracle** sampling. We test them on the most challenging dataset, COCO Medium, and report the mean and standard deviation for the evaluation metrics across 5 runs.

As shown in Table 2, greedy decoding and stochastic sampling, both of which consider states that are likely to be visited by the parametrized policy, outperform the oracle sampling, which only considers states on optimal trajectories. This is particularly apparent in the F1 score, which can be increased even after visiting a state that is not on an optimal trajectory. The results are consistent with the theory from [20, 1]. The performance difference between the first two strategies was not significant, so from here on we choose the simpler method, greedy decoding, when training a model with the proposed multiset loss function.

## 5.5 Experiment 3: Loss Function Comparison

We now compare the proposed multiset loss function against the five baseline loss functions: reinforcement learning $\mathcal{L}_{\text{RL}}$, aggregate distribution matching–$\mathcal{L}_{\text{dm}}^{1}$ and $\mathcal{L}_{\text{dm}}^{\text{KL}}$–, its one-step variant $\mathcal{L}_{\text{1-step}}$, and sequence prediction $\mathcal{L}_{\text{seq}}$, introduced in Section 3. Refer to Appendix F for additional details.

**MNIST Multi** We present the results on the MNIST Multi variants in Table 3 (a). On all three variants and according to both metrics, the proposed multiset loss function outperforms all the others. The reinforcement learning based approach closely follows behind. Its performance, however, drops as the number of items in a target multiset increases. This is understandable, as the variance of policy gradient grows as the length of an episode grows. A similar behaviour was observed with sequence prediction as well as aggregate distribution matching. We were not able to train any decent models with the one-step variant of aggregate distribution matching. This was true especially in terms of exact match (EM), which we attribute to the one-step variant not being capable of modelling dependencies among the predicted items.

**MS COCO** Similar to the results on the variants of MNIST Multi, the proposed multiset loss function matches or outperforms all the others on the two variants of MS COCO, as presented in Table 3 (b). On COCO Easy, with only two objects to predict per example, both aggregated distribution matching (with KL divergence) and the sequence loss functions are as competitive as the proposed multiset loss. The other loss functions significantly underperform these three loss functions, as they did on MNIST Multi. The performance gap between the proposed loss and the others, however, grows substantially on the more challenging COCO Medium, which has more objects per example. The proposed multiset loss outperforms the aggregated distribution matching with KL divergence by 3.7 percentage points on exact match and 4.8 on F1. This is analogous to the experiments on MNIST Multi, where the performance gap increased when moving from four to ten digits.

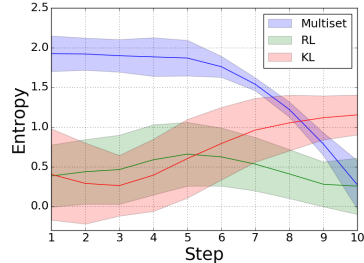

### 5.6 Analysis: Entropy Evolution

Recall from Remark 4 that the entropy of the oracle policy's predictive distribution strictly decreases over time, i.e., $\mathcal{H}(\pi_*^{(t)}) > \mathcal{H}(\pi_*^{(t+1)})$. This naturally follows from the fact that there is no pre-specified rank function, because the oracle policy cannot prefer any item from the others in a free label multiset. Hence, we examine here how the policy learned based on each loss function compares to the oracle policy in terms of per-step entropy. We consider the policies trained on MNIST Multi (10), where the differences among them

Figure 1: Comparison of per-step entropies of predictive distributions compared over the validation set.

were most clear. As shown in Fig. 1, the policy trained on MNIST Multi (10) using the proposed multiset loss closely follows the oracle policy. The entropy decreases as the predictions are made. The decreases can be interpreted as concentrating probability mass on progressively smaller free labels sets. The variance is quite small, indicating that this strategy is uniformly applied for any input.

The policy trained with reinforcement learning retains a relatively low entropy across steps, with a decreasing trend in the second half. We carefully suspect the low entropy in the earlier steps is due to the greedy nature of policy gradient. The policy receives a high reward more easily by choosing one of many possible choices in an earlier step than in a later step. This effectively discourages the policy from exploring all possible trajectories during training.

On the other hand, the policy found by aggregated distribution matching ($\mathcal{L}_{\text{dm}}^{\text{KL}}$) has the opposite behaviour. The entropy in general grows as more predictions are made. To see why this is sub-optimal, consider the final step. Assuming the first nine predictions were correct, there is only one correct class left for the final prediction . The high entropy, however, indicates that the model is placing a significant amount of probability on incorrect sequences. Such a policy may result because $\mathcal{L}_{\text{dm}}^{\text{KL}}$ cannot properly distinguish between policies with increasing and decreasing entropies. The increasing entropy also indicates that the policy has learned a rank function implicitly and is fully relying on it. We conjecture this reliance on an inferred rank function, which is by definition sub-optimal, resulted in lower performance of aggregate distribution matching.

## 6 Conclusion

We have extensively investigated the problem of multiset prediction in this paper. We rigorously defined the problem, and proposed to approach it from the perspective of sequential decision making. In doing so, an oracle policy was defined and shown to be optimal, and a new loss function, called *multiset loss*, was introduced as a means to train a parametrized policy for multiset prediction. The experiments on two families of datasets, MNIST Multi variants and MS COCO variants, have revealed the effectiveness of the proposed loss function over other loss functions including reinforcement learning, sequence, and aggregated distribution matching loss functions. This success brings in new opportunities of applying machine learning to various new domains, including high-energy physics.

### Acknowledgments

KC thanks support by eBay, TenCent, NVIDIA and CIFAR. This work was supported by Samsung Electronics (Improving Deep Learning using Latent Structure) and 17JC1404101 STCSM.

## Footnotes

[1]A set that allows multiple instances, e.g. $\{x, y, x\}$. See Appendix A for a detailed definition.

[2]The number of all possible multisets of size $\leq K$ is $\sum_{k=1}^K \frac{(|C|+k-1)!}{k!(|C|-1)!}$.

[3]The random rank function is generated before training and held fixed. We verified that generating a new random rank function for each batch significantly decreased performance.

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
