[Supplementary Material]

# Appendix: Loss Functions for Multiset Prediction

**Sean Welleck**[1,2], **Zixin Yao**[1], **Yu Gai**[1], **Jialin Mao**[1], **Zheng Zhang**[1], **Kyunghyun Cho**[2,3]

[1]New York University Shanghai
[2]New York University
[3]CIFAR Azrieli Global Scholar
{wellecks,zixin.yao,yg1246,jialin.mao,zz,kyunghyun.cho}@nyu.edu

## A  Definitions

We review definitions of multiset and exact match, and present multiset versions of precision, recall, and F1. For a comprehensive overview of multisets, refer to [5, 3].

**Multiset**   A *multiset* is a set that allows for multiple instances of elements. Multisets are unordered, i.e. $\{x, x, y\}$ and $\{x, y, x\}$ are equal. We now introduce the formal definition and convenient ways of representing a multiset.

Formally, a multiset is a pair $\mathcal{Y} = (C, \mu)$, where $C = \{c_1, ..., c_p\}$ is a *ground set*, and $\mu : C \to \mathbb{N}_{\geq 0}$ is a *multiplicity function* that maps each $c_i \in C$ to the number of times it occurs in the multiset. The multiset cardinality is defined as $|\mathcal{Y}| = \sum_{c \in C} \mu(c)$.

A multiset can be *enumerated* by numbering each element instance and representing the multiset as a size $|\mathcal{Y}|$ set: $\mathcal{Y} = \{c_1^{(1)}, c_1^{(2)}, ..., c_1^{(\mu(c_1))}, c_2^{(1)}, ..., c_2^{(\mu(c_2))}, ..., c_p^1, ..., c_p^{(\mu(c_p))}\}$. This allows for notation such as $\sum_{c \in \mathcal{Y}}$.

An additional compact notation is $\mathcal{Y} = \{y_1, y_2, ..., y_{|\mathcal{Y}|}\}$, where each $y_i$ is an auxiliary variable referring to an underlying element $c \in C$ of the ground set.

For instance, the multiset $\mathcal{Y} = \{\text{cat}, \text{cat}, \text{dog}\}$ can be defined as $\mathcal{Y} = (C, \mu)$, where $C = \{c_1 = \text{cat}, c_2 = \text{dog}, c_3 = \text{fish}\}$, $\mu(\text{cat}) = 2, \mu(\text{dog}) = 1, \mu(\text{fish}) = 0$, and can be written as $\mathcal{Y} = \{c_1^{(1)} = \text{cat}, c_1^{(2)} = \text{cat}, c_2^{(1)} = \text{dog}\}$ or $\mathcal{Y} = \{y_1 = \text{cat}, y_2 = \text{cat}, y_3 = \text{dog}\}$.

For multiset analogues of common set operations (e.g. union, intersection, difference), and the notion of a subset, see [5, 3].

**Exact Match (EM)**   Two multisets *exactly match* when their elements and multiplicities are the same. For example, $\{x, y, x\}$ exactly matches $\{y, x, x\}$, while $\{x, y, x\}$ does not exactly match $\{z, y, z\}$ or $\{x, y\}$.

Formally, let $\hat{\mathcal{Y}} = (\mathcal{C}, \mu_{\hat{Y}})$, $\mathcal{Y} = (\mathcal{C}, \mu_Y)$ be multisets over a common ground set $\mathcal{C}$. Then $\hat{\mathcal{Y}}$ and $\mathcal{Y}$ exactly match if and only if $\mu_{\hat{Y}}(c) = \mu_Y(c)$ for all $c \in \mathcal{C}$. The evaluation metric $\text{EM}(\hat{\mathcal{Y}}, \mathcal{Y})$ is 1 when $\hat{\mathcal{Y}}$ and $\mathcal{Y}$ exactly match, and 0 otherwise.

Note that exact match is the same as multiset equality, i.e. $\hat{\mathcal{Y}} = \mathcal{Y}$, as defined in [3].

**Precision**   Precision gives the ratio of correctly predicted elements to the number of predicted elements. Specifically, let $\hat{\mathcal{Y}} = (\mathcal{C}, \mu_{\hat{Y}})$, $\mathcal{Y} = (\mathcal{C}, \mu_Y)$ be multisets. Then

$$\text{Prec}(\hat{\mathcal{Y}}, \mathcal{Y}) = \frac{\sum_{y \in \hat{\mathcal{Y}}} I_{y \in \mathcal{Y}}}{|\hat{\mathcal{Y}}|}.$$

The summation and membership are done by enumerating the multiset. For example, the multisets $\hat{\mathcal{Y}} = \{a, a, b\}$ and $\mathcal{Y} = \{a, b\}$ are enumerated as $\hat{\mathcal{Y}} = \{a^{(1)}, a^{(2)}, b^{(1)}\}$ and $\mathcal{Y} = \{a^{(1)}, b^{(1)}\}$, respectively. Then clearly $a^{(1)} \in \mathcal{Y}$ but $a^{(2)} \notin \mathcal{Y}$.

Formally, precision can be defined as

$$\text{Prec}(\hat{\mathcal{Y}}, \mathcal{Y}) = 1 - \frac{\sum_{c \in \mathcal{C}} \max\left(\mu_{\hat{Y}}(c) - \mu_Y(c), 0\right)}{|\hat{\mathcal{Y}}|}$$

where the summation is now over the ground set $\mathcal{C}$. Intuitively, precision decreases by $\frac{1}{|\hat{\mathcal{Y}}|}$ each time an extra class label is predicted.

**Recall**  Recall gives the ratio of correctly predicted elements to the number of ground-truth elements. Recall is defined analogously to precision, as:

$$\text{Rec}(\hat{\mathcal{Y}}, \mathcal{Y}) = \frac{\sum_{y \in \hat{y}} I_{y \in \mathcal{Y}}}{|\mathcal{Y}|}.$$

Formally,

$$\text{Rec}(\hat{\mathcal{Y}}, \mathcal{Y}) = 1 - \frac{\sum_{c \in \mathcal{C}} \max\left(\mu_Y(c) - \mu_{\hat{Y}}(c), 0\right)}{|\mathcal{Y}|}.$$

Intuitively, recall decreases by $\frac{1}{|\mathcal{Y}|}$ each time an element of $\mathcal{Y}$ is not predicted.

**F1**  The F1 score is the harmonic mean of precision and recall:

$$\text{F}_1(\hat{\mathcal{Y}}, \mathcal{Y}) = 2 \cdot \frac{\text{Prec}(\hat{\mathcal{Y}}, \mathcal{Y}) \cdot \text{Rec}(\hat{\mathcal{Y}}, \mathcal{Y})}{\text{Prec}(\hat{\mathcal{Y}}, \mathcal{Y}) + \text{Rec}(\hat{\mathcal{Y}}, \mathcal{Y})}.$$

# B  Proof of Remark 1

*Proof.* Note that the precision with $\hat{y}_{<t}$ is defined as
$$\text{Prec}(\hat{y}_{<t}, \mathcal{Y}) = \frac{\sum_{y \in \hat{y}_{<t}} I_{y \in \mathcal{Y}}}{|\hat{y}_{<t}|}.$$
Because $\hat{y} \sim \pi_*(\hat{y}_{<t}, x, \mathcal{Y}_t) \in \mathcal{Y}_t$,
$$\text{Prec}(\hat{y}_{\leq t}, \mathcal{Y}) = \frac{1 + \sum_{y \in \hat{y}_{<t}} I_{y \in \mathcal{Y}}}{1 + |\hat{y}_{<t}|}.$$
Then,
$$\text{Prec}(\hat{y}_{\leq t}, \mathcal{Y}) - \text{Prec}(\hat{y}_{<t}, \mathcal{Y}) = \frac{1 - \text{Prec}(\hat{y}_{<t}, \mathcal{Y})}{1 + |\hat{y}_{<t}|} \geq 0,$$
because $0 \leq \text{Prec}(\hat{y}_{<t}, \mathcal{Y}) \leq 1$ and $|\hat{y}_{<t}| \geq 0$. The equality holds when $\text{Prec}(\hat{y}_{<t}, \mathcal{Y}) = 1$.

Similarly, we start with the definition of the recall:
$$\text{Rec}(\hat{y}_{<t}, \mathcal{Y}) = \frac{\sum_{y \in \hat{y}_{<t}} I_{y \in \mathcal{Y}}}{|\mathcal{Y}|}.$$
Because $\hat{y} \sim \pi_*(\hat{y}_{<t}, x, \mathcal{Y}_t) \in \mathcal{Y}_t$,
$$\text{Rec}(\hat{y}_{\leq t}, \mathcal{Y}) = \frac{1 + \sum_{y \in \hat{y}_{<t}} I_{y \in \mathcal{Y}}}{|\mathcal{Y}|}.$$
Since the denominator is identical,
$$\text{Rec}(\hat{y}_{\leq t}, \mathcal{Y}) - \text{Rec}(\hat{y}_{<t}, \mathcal{Y}) = \frac{1}{|\mathcal{Y}|} \geq 0.$$

$\square$

## C   Proof of Remark 2

*Proof.* When $t = 1$,
$$\text{Prec}(\hat{y}_{\leq 1}, \mathcal{Y}) = 1,$$
because $\hat{y}_1 \sim \pi_*(\emptyset, x, \mathcal{Y}_1) \in \mathcal{Y}$. From Remark 1, we know that
$$\text{Prec}(\hat{y}_{\leq t}, \mathcal{Y}) = \text{Prec}(\hat{y}_{<t}, \mathcal{Y}),$$
when $\text{Prec}(\hat{y}_{<t}, \mathcal{Y}) = 1$. By induction, $\text{Prec}(\hat{y}_{\leq|\mathcal{Y}|}, \mathcal{Y}) = 1$.

From the proof of Remark 1, we know that the recall increases by $\frac{1}{\mathcal{Y}}$ each time, and we also know that
$$\text{Rec}(\hat{y}_{\leq 1}, \mathcal{Y}) = \frac{1}{|\mathcal{Y}|},$$
when $t = 1$. After $|\mathcal{Y}| - 1$ steps of executing the oracle policy, the recall becomes

$$\text{Rec}(\hat{y}_{\leq|\mathcal{Y}|}, \mathcal{Y}) = \frac{1}{|\mathcal{Y}|} + \sum_{t'=2}^{|\mathcal{Y}|} \frac{1}{|\mathcal{Y}|} = 1.$$

$\square$

## D   Proof of Remark 4

*Proof.* Given a multiset $\mathcal{Y}$ with $|\mathcal{Y}| \leq M$, define $\mathbf{C} = \{c_i^{(m)} | 1 \leq i \leq |\mathcal{C}|, 1 \leq m \leq M\}$, where $c_i^{(m)}$ is interpreted as the $m$'th instance of class $c_i$. Writing $\mathcal{Y}$ in enumerated form it is clear that $\mathcal{Y} \subset \mathbf{C}$. Let $t$ range from 1 to $|\mathcal{Y}|$ and define $\mathcal{Y}_t$ as in Definition 1.

Now, define the oracle policy as a distribution over $\mathbf{C}$, according to Definition 2:

$$\pi_*^{(t)}(y = c_i^{(m)} | \hat{y}_{<t}, x, \mathcal{Y}_t) = \left\{ \begin{array}{ll} \frac{1}{|\mathcal{Y}_t|}, & \text{if } c_i^{(m)} \in \mathcal{Y}_t \\ 0, & \text{otherwise} \end{array} \right. .$$

Therefore,

$$\begin{aligned}
\mathcal{H}\left(\pi_*^{(t)}\right) &= -\sum_{i=1}^{|\mathcal{C}|} \sum_{m=1}^{M} \pi_*^{(t)}(y = c_i^{(m)}) \log \pi_*^{(t)}(y = c_i^{(m)}) \\
&= -\sum_{c \in \mathcal{Y}_t} \frac{1}{|\mathcal{Y}_t|} \log \frac{1}{|\mathcal{Y}_t|} \\
&= \frac{1}{|\mathcal{Y}_t|} \sum_{c \in \mathcal{Y}_t} \log |\mathcal{Y}_t| \\
&= \log |\mathcal{Y}_t|
\end{aligned}$$

where $0 \log 0$ is defined as 0 in the first step.

Now, observe that $|\mathcal{Y}_t| > |\mathcal{Y}_{t+1}|$ since $\hat{y}_t \sim \pi_*^{(t)}$ is in $\mathcal{Y}_t$ with probability 1 and $\mathcal{Y}_{t+1} \leftarrow \mathcal{Y}_t \backslash \{\hat{y}_t\}$ by definition. Hence
$$\mathcal{H}\left(\pi_*^{(t)}\right) = \log |\mathcal{Y}_t| > \log |\mathcal{Y}_{t+1}| = \mathcal{H}\left(\pi_*^{(t+1)}\right).$$

$\square$

## E   Model Descriptions

**Model**   An input $x$ is first processed by a tower of convolutional layers, resulting in a feature volume of size $w' \times h'$ with $d$ feature maps, i.e., $H = \phi(x) \in \mathbb{R}^{w' \times h' \times d}$. At each time step $t$, we resize the previous prediction's embedding $\text{emb}(\hat{y}_{t-1}) \in \mathbb{R}^{(w')(h')}$ to be a $w' \times h'$ tensor and concatenate it with $H$, resulting in $\tilde{H} \in \mathbb{R}^{w' \times h' \times (d+1)}$. This feature volume is then fed into a stack of convolutional

Figure 1: Graphical illustration of a predictor used throughout the experiments.

LSTM layers. The output from the final convolutional LSTM layer $C \in \mathbb{R}^{w' \times h' \times q}$ is spatially average-pooled, i.e., $c = \frac{1}{w'h'} \sum_{i=1}^{w'} \sum_{j=1}^{h'} C_{i,j,\cdot} \in \mathbb{R}^q$. This feature vector $c$ is then turned into a conditional distribution over the next item after affine transformation followed by a softmax function. When the one-step variant of aggregated distribution matching is used, we skip the convolutional LSTM layers, i.e., $c = \frac{1}{w'h'} \sum_{i=1}^{w'} \sum_{j=1}^{h'} H_{i,j,\cdot} \in \mathbb{R}^d$.

See Fig. 1 for the graphical illustration of the entire network. See Table 1 for the details of the network for each dataset.

**Preprocessing** For MNIST Multi, we do not preprocess the input at all. In the case of MS COCO, input images are of different sizes. Each image is first resized so that its larger dimension has 600 pixels, then along its other dimension is zero-padded to 600 pixels and centered, resulting in a 600x600 image.

**Training** The model is trained end-to-end, except ResNet-34 which remains fixed after being pretrained on ImageNet. For all the experiments, we train a neural network using Adam [2] with a fixed learning rate of 0.001, $\beta$ of (0.9, 0.999) and $\epsilon$ of 1e-8. The learning rate was selected based on the validation performance during the preliminary experiments, and the other parameters are the default values. For MNIST Multi, the batch size was 64, and for COCO was 32. For the selection strategy experiments, 5 runs with different random seeds were used.

Table 1: Network Architectures

| Data | MNIST Multi | MS COCO |
|---|---|---|
| CNN | conv $5 \times 5$ feat 10<br>max-pool $2 \times 2$<br>conv $5 \times 5$ feat 10<br>max-pool $2 \times 2$<br>conv $5 \times 5$ feat 32<br>max-pool $2 \times 2$ | ResNet-34 |
| emb($\hat{y}_{t-1}$) | 81 | 361 |
| ConvLSTM | conv $3 \times 3$ feat 32<br>conv $3 \times 3$ feat 32 | conv $3 \times 3$ feat 512<br>conv $3 \times 3$ feat 512 |

**Feedforward Alternative** While we use a recurrent model in the experiments, the multiset loss can be used with a feedforward model as follows. A key use of the recurrent hidden state is to retain the previously predicted labels, i.e. to remember the full conditioning set $\hat{y}_1, ..., \hat{y}_{t-1}$ in $p(y_t|\hat{y}_1, ..., \hat{y}_{t-1})$. Therefore, the proposed loss can be used in a feedforward model by encoding $\hat{y}_1, ..., \hat{y}_{t-1}$ in the input $x_t$, and running the feedforward model for $|\hat{\mathcal{Y}}|$ steps, where $|\hat{\mathcal{Y}}|$ is determined using a termination policy or the Special Class method detailed below. Note that compared to the recurrent model, this approach involves additional feature engineering.

**Termination Policy Alternative: Special Class** An alternative strategy to support predicting variable-sized multisets is to introduce a special item to the class set, called $\langle \text{END} \rangle$, and add it to the

final free label multiset $\mathcal{Y}_{|\mathcal{Y}|+1} = \{\langle \text{END} \rangle\}$. Thus, the parametrized policy is trained to predict this special item $\langle \text{END} \rangle$ once all the items in the target multiset have been predicted. This is analogous to NLP sequence models which predict an end of sentence token [4, 1], and was used in [6] to predict variable-sized multisets.

# F    Additional Experimental Details

## F.1    Baseline Loss Functions

### F.1.1    $\mathcal{L}_{\text{1-step}}$

The corresponding loss function for the one-step distribution matching baseline introduced in 3.1.1, $\mathcal{L}_{\text{1-step}}$, is:

$$\mathcal{L}_{\text{1-step}}(x, \mathcal{Y}, \theta) = \sum_{c \in C} q_*(c|x) \log q_\theta(c|x) + \lambda(\hat{l}_\theta(x) - |\mathcal{Y}|)^2,$$

where $\lambda > 0$ is a coefficient for balancing the contributions from the two terms.

### F.1.2    $\mathcal{L}_{\text{seq}}$

First define a rank function $r$ that maps from one of the unique items in the class set $c \in C$ to a unique integer. That is, $r : C \to \mathbb{Z}$. This function assigns the rank of each item and is used to order items $y_i$ in a target multiset $\mathcal{Y}$. This results in a sequence $\mathcal{S} = (s_1, \ldots, s_{|\mathcal{Y}|})$, where $r(s_i) \geq r(s_j)$ for all $j > i$, and $s_i \in \mathcal{Y}$.

With this target sequence $\mathcal{S}$ created from $\mathcal{Y}$ using the rank function $r$, the sequence loss function is defined as

$$\mathcal{L}_{\text{seq}}(x, \mathcal{S}, \theta) = -\sum_{t=1}^{|\mathcal{S}|} \log \pi_\theta(s_t | s_{<t}, x).$$

Minimizing this loss function is equivalent to maximizing the conditional log-probability of the sequence $\mathcal{S}$ given $x$.

### F.1.3    $\mathcal{L}_{\text{dm}}$

In distribution matching, we consider the target multiset $\mathcal{Y}$ as a set of samples from a single, underlying distribution $q^*$ over the class set $C$. This underlying distribution can be empirically estimated by counting the number of occurrences of each item $c \in C$ in $\mathcal{Y}$. That is,

$$q_*(c|x) = \frac{1}{|\mathcal{Y}|} \sum_{y \in \mathcal{Y}} I_{y=c},$$

where $I$ is the indicator function.

Similarly, we can construct an aggregated distribution computed by the parametrized policy, here denoted as $q_\theta(c|x)$. To do so, the policy predicts $(y_1, ..., y_{|\mathcal{Y}|})$ by sampling from a predicted distribution $q_\theta^{(t)}(y_t|y_{<t}, x)$ at each step $t$. The per-step distributions $q_\theta^{(t)}$ are then averaged to form the aggregate distribution $q_\theta$.

Learning is equivalent to minimizing a divergence between $q_*$ and $q_\theta$. The $\mathcal{L}_{\text{dm}}^p$ baseline uses

$$\mathcal{L}_{\text{dm}}^p(x, \mathcal{Y}, \theta) = \|q_* - q_\theta\|_p,$$

where $q_*$ and $q$ are the vectors representing the corresponding categorical distributions, and $p = 1$ in the experiments. The $\mathcal{L}_{\text{dm}}^{\text{KL}}$ baseline uses KL divergence:

$$\mathcal{L}_{\text{dm}}^{\text{KL}}(x, \mathcal{Y}, \theta) = -\sum_{c \in C} q_*(c|x) \log q_\theta(c|x).$$

### F.1.4 $\mathcal{L}_{\mathrm{RL}}$

Instead of assuming the existence of an oracle policy, this approach solely relies on a reward function $r$ designed specifically for multiset prediction. The reward function is defined as

$$r(\hat{y}_t, \mathcal{Y}_t) = \left\{ \begin{array}{ll} 1, & \text{if } \hat{y}_t \in \mathcal{Y}_t \\ -1, & \text{otherwise} \end{array} \right.$$

The goal is then to maximize the sum of rewards over a trajectory of predictions from a parametrized policy $\pi_\theta$. The final loss function is

$$\mathcal{L}_{\mathrm{RL}} = -\mathbb{E}_{\hat{y} \sim \pi_\theta} \left[ \sum_{t=1}^{T} r(\hat{y}_{<t}, \mathcal{Y}_t) - \lambda \mathcal{H}(\pi_\theta(\hat{y}_{<t}, x)) \right] \tag{1}$$

where the second term inside the expectation is the negative entropy multiplied with a regularization coefficient $\lambda$. The second term encourages exploration during training. As in [6], we use REINFORCE [7] to stochastically minimize the loss function above with respect to $\pi_\theta$. This loss function is optimal in that the return, i.e., the sum of the step-wise rewards, is maximized when both the precision and recall are maximal ($= 1$).

## F.2 Input-Dependent Rank Function

For the $\mathcal{L}_{\mathrm{seq}}$ baseline, a domain-specific, input-dependent rank function can be defined to transform the target multiset into a sequence. A representative example is an image input with bounding box annotations. Here, we present two input-dependent rank functions in such a case.

First, a spatial rank function $r_{\mathrm{spatial}}$ assigns an integer rank to each item in a given target multiset $\mathcal{Y}$ such that

$$r_{\mathrm{spatial}}(y_i|x) < r_{\mathrm{spatial}}(y_j|x),$$
$$\text{if } \mathrm{pos}_x(x_i) < \mathrm{pos}_x(x_j) \text{ and } \mathrm{pos}_y(x_i) < \mathrm{pos}_y(x_j),$$

where $x_i$ and $x_j$ are the objects corresponding to the items $y_i$ and $y_j$.

Second, an area rank function $r_{\mathrm{area}}$ decides the rank of each label in a target multiset according to the size of the corresponding object inside the input image:

$$r_{\mathrm{area}}(y_i|x) < r_{\mathrm{area}}(y_j|x), \text{ if } \mathrm{area}(x_i) < \mathrm{area}(x_j).$$

The area may be determined based on the size of a bounding box or the number of pixels, depending on the level of annotation.