[Reviews · NeurIPS 2018]

Reviewer 1



This paper studies the problem of multiset prediction, where the task in to predict a multiset of labels out of the set of allowed multisets. The proposed method does sequential predictions of labels and is trained to imitate the optimal oracle strategy. The method is evaluated on the two tasks: MultiMNIST and recognition of multiple objects on the COCO dataset. The paper is clearly written, explains the method and some theoretical properties well. The description of the experiments looks good enough. 1. As explained in Lines 191-197 the proposed method is essentially an instantiation of the learning to search approach [2, 1, 10]. However the method is run in a regime, which was considered suboptimal in the previous studies (learned roll-in and reference roll-out). In particular, [1] and [10] concluded that one should do learned roll-in and mixed (reference and learned) roll-outs. Comparisons to different versions of the method are important to understand the benefits of the method. 2. It is very hard to judge how important the task of multiset prediction is. The experiments in the paper in toy settings (I'm not saying that those are easy to solve). Lines 16-19 give some contexts where the setting could be useful, but there are no attempts to apply the method in any of those. 3. The proposed loss based on KL is not really novel. It is very similar to the KL loss used in [10] and in [Norouzi et al., Reward Augmented Maximum Likelihood for Neural Structured Prediction, NIPS 2016]. 4. Table 3 (the paper comparing the baselines) of the paper does not have error bars, so it not clear how stable the results are. === after the response I've read the response and would like to thank the authors for reply on question 1. However, I still have concerns that the contribution of the paper is relatively incremental and the task is not proven to be significant (it is mentioned that multiset prediction can be used as a proxy for other tasks, but I'd really want to see examples of that) so I'm keeping my old score.

Reviewer 2



The paper is proposing a interesting approach for multiset prediction and compares against many other baselines. The approach seems simple to implement with any software that can compute gradients. The authors should consider including a discussion on the cicumstances under which the proposed loss approximates well/badly the per step loss for every possible state.

Reviewer 3



The paper proposes a novel method for multiset prediction, which is to predict, for each class, the number of instances of that class present in the example. As in previous work, the problem is reduced to training a policy to predict a possibly-repeating sequence of labels. The main novel proposal in this work is to train the policy to minimize the KL divergence between the expert policy and the model policy, which avoids the need to define an arbitrary order in which to predict the sequence (as in other sequence-based methods), and also sidesteps the optimization difficulties of the RL-based approach. Experiments show that this method substantially outperforms existing methods on the task of multiple-instance prediction for images. Pros: 1. The proposed method is simple, well-motivated and carefully designed, and performs well in practice 2. The task seems like a good abstraction for a number of practical applications, and the proposed method solves it very well 3. Focused experiments systematically motivate the method, validate hypotheses, and demonstrate the real-world usefulness and substantial advantages of the method over existing methods 4. Paper is clearly written overall Cons: 1. The novelty and generality of the technique deployed here may be a bit on the low side 2. One relevant baseline may be missing: using a graphical model over the instance counts of different classes Overall, I believe the paper makes a compelling case for the proposed method. It is simple and substantially outperforms what seem to be reasonably strong baselines for the task on real-world problems. The method is well-motivated, and the paper does a good job of articulating the careful consideration that went into designing the method, and why it might be expected to outperform existing approaches. The experiments are also well-designed, focusing on answering the most important and interesting questions. I could also imagine this work having significant impact, because many interesting tasks could be formulated in terms of multiset prediction. I believe the paper does a commendable job of categorizing the different possible approaches to this problem, explaining the advantages of the proposed method, and empirically demonstrating those advantages. The proposed method avoids the arbitrary-permutation-dependence of some other sequential prediction approaches, and avoids having to do RL over a long time horizon, as in RL-based methods. I appreciated how these advantages were clearly articulated and then validated experimentally with targeted experiments. I did have a doubt with regard to the “one-step distribution matching” baseline—this was tried in the experiments, and the paper cited the “lack of modeling of dependencies among the items” as a potential explanation for its poor performance, but I found this a bit unsatisfying. I think it should be possible to define a graphical model where each node counts the number of instances of a particular class, and cliques model correlations between counts of different classes. The node and clique potentials could then be learned so as to maximize the data likelihood. Since this model may be non-trivial, and I do not have a citation for it, I do not see the lack of this baseline as an egregious oversight, however. I also noticed that although the lack of a need to perform RL is claimed as a benefit of this method over [25], I think you could argue that the proposed method could be thought of as an approximation of a more natural method that would also require RL. Specifically, the “most natural” version of the multiset loss function would be marginalized over all possible policy roll-outs. However, it seems that in practice, the method just samples one roll-out, and optimizes the resulting loss with that roll-out fixed. The resulting loss gradient is therefore biased, because it does not take into account how the sampled roll-out changes as the policy parameters change. To un-bias the gradient, one would again have to resort to policy gradient. If it turns out that neglecting the bias is fine in practice, then that is fine; however, it should probably be acknowledged. On the other hand, I would like to point out that the choice to sample past states from the policy instead of the expert is a subtle but smart choice, and the paper both includes a citation as to why this is superior, and demonstrates empirically that this choice indeed outperforms the naive approach of sampling past states from the expert when evaluating the loss. In terms of novelty, I would say that might be one of the weaker aspects of the work, since this is technically a rather straightforward method. However, I think there is nonetheless significance in carefully and systematically designing a simple method to perform well for a given task, especially taking subtle considerations such as that mentioned above into account. In summary, I think the clarity and simplicity of the method, combined with the compelling experimental results shown here, will make this work interesting to a significant cross-section of the NIPS audience. POST-REBUTTAL COMMENTS I think the paper should acknowledge the connection to previous methods (mentioned by reviewer 1) that employed a KL loss for similar problems. Mentioning the potential applications to computer vision problems up-front may also help motivate the problem better, because 'multisets' may sound too abstract to many readers. As for the issue of potential bias in the gradient, I am still a little unclear on this. It seems like there are two ways to interpret this method: as an RL method, or as an online learning problem (e.g., learning to search). I am more familiar with the former, but I vaguely understand that there is a regret-based analysis where the bias issue I brought up doesn't directly factor in, or may appear in some other guise. Ideally, it would also be nice to have some clarity on this issue, though it may be too tangential for this submission.